# Myo/Nog Cells: The Jekylls and Hydes of the Lens

**DOI:** 10.3390/cells12131725

**Published:** 2023-06-27

**Authors:** Jacquelyn Gerhart, Mindy George-Weinstein

**Affiliations:** Division of Research, Philadelphia College of Osteopathic Medicine, Philadelphia, PA 19131, USA; jacquelynge@pcom.edu

**Keywords:** Myo/Nog, MyoD, Noggin, BAI1, myofibroblasts, fibrosis, lens, retina, PCO, PVR

## Abstract

Herein, we review a unique and versatile lineage composed of Myo/Nog cells that may be beneficial or detrimental depending on their environment and nature of the pathological stimuli they are exposed to. While we will focus on the lens, related Myo/Nog cell behaviors and functions in other tissues are integrated into the narrative of our research that spans over three decades, examines multiple species and progresses from early stages of embryonic development to aging adults. Myo/Nog cells were discovered in the embryonic epiblast by their co-expression of the skeletal muscle-specific transcription factor MyoD, the bone morphogenetic protein inhibitor Noggin and brain-specific angiogenesis inhibitor 1. They were tracked from the epiblast into the developing lens, revealing heterogeneity of cell types within this structure. Depletion of Myo/Nog cells in the epiblast results in eye malformations arising from the absence of Noggin. In the adult lens, Myo/Nog cells are the source of myofibroblasts whose contractions produce wrinkles in the capsule. Eliminating this population within the rabbit lens during cataract surgery reduces posterior capsule opacification to below clinically significant levels. Parallels are drawn between the therapeutic potential of targeting Myo/Nog cells to prevent fibrotic disease in the lens and other ocular tissues.

## 1. Discovery of Myo/Nog Cells in the Embryo

Myo/Nog cells were discovered during our search for the origin of the skeletal muscle lineage, which, at the time, was reported to occur in the embryonic somites under the influence of factors released by surrounding tissues [1,2,3,4]. A few outliers to this well documented induction of cell fate demonstrated that the tissues which give rise to the somites, including epiblast, contain cells that differentiate into skeletal muscle in culture, even when grown in serum-free medium [5,6,7,8]. In vivo, we detected approximately 14 cells in the epiblast of the *chick* embryo blastocyst and 70 cells by the onset of gastrulation that expressed mRNA for the skeletal muscle-specific transcription factor MyoD (Figure 1B) [9] that, along with other members of its family, regulate the specification of cells to the skeletal muscle lineage [10,11,12].

Fortuitously, a monoclonal antibody (mAb) we generated against mesoderm cells, named G8, specifically bound to cells with MyoD mRNA in the epiblast (Figure 1B) and fetal organs lacking skeletal muscle [13,14,15]. The antigen recognized by the G8 mAb was later identified as brain-specific angiogenesis inhibitor 1 (BAI1) [16], a member of the adhesion G protein-coupled receptor family [17,18]. Functions of BAI1 include, among others, inhibition of angiogenesis and mediation of myocyte fusion to form multinucleated myofibers and phagocytosis [17,18,19,20,21,22,23,24]. The G8 epitope lies within the third thrombospondin repeat of BAI1′s extracellular domain [16], thereby enabling us to use the G8 mAb to isolate, track and target MyoD-positive (+) cells.

BAI1+/MyoD+ cells isolated from the *chick* embryo epiblast with the G8 mAb differentiated into skeletal muscle and fused to form multinucleated myofibers with sarcomeres when cultured in serum-free medium [14] (Figure 1D). BAI1-negative (−) cells differentiated predominantly into cardiac muscle (Figure 1E) unless they were cultured in medium conditioned by BAI1+ cells in which they underwent skeletal myogenesis [14]. We speculated that the factor(s) released by the BAI1+ cells in vitro was related to the mechanism regulating myogenesis in the somites involving inhibition of bone morphogenetic protein (BMP) signaling via Noggin [25,26,27,28]. This hypothesis was validated when skeletal myogenesis was induced in BAI1- cultures by substituting BAI1+ cell-conditioned medium with Noggin [14] (Figure 1F). Noggin mRNA and protein were then localized exclusively in epiblast cells that expressed MyoD and BAI1 in vivo (Figure 1C). The triple-positive cells were named Myo/Nog to reflect their capacity to differentiate into skeletal muscle and release Noggin. 

The anti-BAI1 G8 mAb was also used to track Myo/Nog cells from the chick embryo epiblast into all three germ layers and their derivatives [29,30,31]. Only a subpopulation of cells in the somites expressed BAI1+/Noggin+ [29], suggesting that Myo/Nog cells are a separate lineage from myogenic cells induced within the mesoderm. BAI1+ epiblast cells were also tracked into the pre-lens placode and lens vesicle, and later were localized in the equatorial and germinative zones, anterior epithelium and between the primary fiber cells [31] (Figure 2A–D). BAI1+ cells continue to express MyoD and Noggin in the lens (Figure 2E,F) and other non-skeletal muscle tissues in the embryo and fetus. Myo/Nog cells were tracked into the heart and brain, and those present in the fetal intestine differentiated into skeletal muscle in culture [13,30]. These studies demonstrated Myo/Nog cells’ stable commitment to the skeletal muscle lineage and capacity for regulating BMP signaling regardless of their environment. Our cell-tracking experiments revealed for the first time that the developing lens consists of two populations, the lens epithelial cell (LEC) lineage and Myo/Nog cells.

## 2. Roles of Myo/Nog Cells in the Developing Embryo

The functions of Myo/Nog cells during development were revealed in a series of experiments in which they were eliminated in the *chick* embryo epiblast by lysing them with the anti-BAI1 G8 mAb and complement [29,31,32]. Once depleted, Myo/Nog cells did not reappear in the embryo and Noggin was not detected in other cell types. The absence of Myo/Nog cells resulted in an expansion of BMP signaling and severe malformations of the ventral body wall, central nervous system and eyes (Figure 3C,D), an absence of skeletal muscle and expansion of cardiac muscle [29,31,32]. These defects phenocopied to a large extent in Noggin null mice [33,34,35,36,37,38]. Eye malformations ranged from anophthalmia to lens dysgenesis and overgrowth of the retina [29,31] (Figure 3C,D). Treatment with the G8 mAb or complement alone did not impair eye development (not shown). Replacing Myo/Nog cells with beads soaked in Noggin prevented ocular defects and other malformations (Figure 3E,F) and restored skeletal muscle differentiation in the somites [29,31]. Thus, Myo/Nog cells’ release of Noggin is indispensable for normal development, including eye morphogenesis. 

Noggin binds to BMPs -2, -4 and -7 and prevents receptor binding [39,40]. Knockout of BMPs and exogenous or overexpressed Noggin have been used as tools to define the roles of BMPs during lens development [41]. BMPs -2, -4 and -7 are important for the specification of the ectoderm to the lens epithelial cell lineage, formation and invagination of the lens placode and lens fiber cell differentiation [42,43,44,45,46,47,48,49,50,51,52]. Although BMPs are required for eye development, the intensity of their signaling must be regulated [53,54,55]. Our studies illustrate that Myo/Nog cells’ release of Noggin is a physiological mechanism that titrates the effects of BMPs in the lens and this is accomplished with a small number of cells [31]. 

Another role for Myo/Nog cells was revealed in a control experiment in which a separate subpopulation of cells in the central epiblast was depleted using the D4 mAb and complement [32]. Elimination of D4+ cells reduced the size of the embryo but the eyes appeared normal. Lysis of D4+ cells stimulated a rapid increase in the number of Myo/Nog cells and their migration from the posterior/medial epiblast to dying cells in the central epiblast [32] (Figure 3G,H). 

Myo/Nog cells also rapidly accumulate around puncture wounds in the embryo (Figure 3J) and incisions made in the fetal lens to produce capsular bag cultures [56,57,58] (Figure 3L). BAI1+/MyoD mRNA+ cells were initially present in niches in the lens explants [59] (Figure 3K). Tracking the cells with the anti-BAI1 G8 mAb demonstrated their rapid migration to the cut edges of the tissue (Figure 3L) and synthesis of MyoD protein and alpha-smooth muscle actin (α-SMA) [59], a marker of vascular smooth muscle cells and myofibroblasts [60]. Lysis of Myo/Nog cells with the G8 mAb and complement prevented the emergence of α-SMA+ cells [59]. These studies of embryonic and fetal chick tissues revealed that Myo/Nog cells home to dying cells and wounds. Myo/Nog cells’ response to injury in an adult mammalian lens and retina will be described below.

## 3. Myo/Nog Cells in the Adult Eye

The next phase of our exploration of Myo/Nog cells involved characterizing their behaviors in normal and diseased tissues of adult mammals. Similar to the *chick* embryo, Myo/Nog cells are present in low numbers in all adult tissues and organs analyzed thus far, from *mice* to *humans* [16,61,62,63,64,65,66,67,68,69,70]. Within the eye, Myo/Nog cells are residents of the lens, cornea, ciliary body, retina and choroid [16,62,63,66,67,68,70,71,72] (Figure 4). They also are present on the zonules of Zinn migrating towards the lens from the ciliary body [64] (Figure 4F). In the adult lens, Myo/Nog cells are associated with LECs in the equatorial, bow and anterior regions and are present but rare among the fiber cells [62,64,66,68] (Figure 4B,C). 

### Behavior of Myo/Nog Cells in Explant Cultures of Anterior Human Lens Tissue

The behaviors of Myo/Nog cells in the adult lens were initially studied in serum-free explant cultures of anterior tissue removed from patients undergoing cataract surgery. MyoD+, Noggin+ and BAI1+ cells were associated with the apical surface of LECs [62,71]. They were also found at the cut edge of the tissue and surrounding wounds within the epithelium where they extended processes towards wrinkles in the capsule (Figure 5A–C). Myo/Nog cells synthesized α-SMA, sarcomeric myosin and skeletal muscle-specific troponin T and the T-tubule-associated 12,101 protein [62] (Figure 5A–C). Only Noggin+/BAI1+ cells were co-labeled with antibodies to striated muscle proteins [62]. The appearance of MyoD, α-SMA and multiple striated muscle proteins defines the differentiated progeny of Myo/Nog cells as myofibroblasts that were shown decades ago to resemble single-nucleated skeletal myocytes [60,73,74,75,76]. 

Differentiating Myo/Nog cells rapidly filled in a wound produced by scratching the epithelium (Figure 5E,F). Depletion of Myo/Nog cells in *human* lens explants with the anti-BAI1 G8 mAb and complement reduced the number of cells populating the scratch wound (Figure 5G) and eliminated cells with MyoD and striated myosin throughout the explant in 5-day cultures [62]. Expression of α-SMA was detected in LECs lacking BAI1 in explants depleted of Myo/Nog cells and treated with transforming growth factor beta 1 (TGF-β1) [62] (Figure 5). TGF-β2, a known inducer of epithelial to mesenchymal transition and expression of α-SMA in the lens [58,77,78,79,80,81,82,83,84,85,86,87,88], was added to explant cultures in an effort to stimulate the migration and transdifferentiation of LECS to myofibroblasts in the absence of Myo/Nog cells. TGF-β2, but not TGF-β1, resulted in the loss of the majority of LECs within 48 h [62] (Figure 5H). Co-addition of Noggin prevented LEC loss, and although the cells filled the wound, they did not synthesize MyoD or sarcomeric myosin [62] (Figure 5I,J). These results suggest that the combination of TGF-β2 and hyperactive BMP signaling is toxic to LECs, and Myo/Nog cells may modulate interactions between the two pathways. Combinatorial effects of these signaling pathways was also revealed in explant cultures of rat lens epithelial cells in which the addition of BMP-7 suppressed TGF-β2′s induction of an epithelial to mesenchymal transition [89].

The long-term effects of Myo/Nog cell depletion in human lens tissue were explored in 30-day serum-free explant cultures [71]. In these experiments, Myo/Nog cells were depleted with the BAI1 mAb conjugated to 3DNA nanocarriers intercalated with doxorubicin (BAI1 Ab:3DNA:Dox) (Figure 5K). The drug was not toxic to surrounding LECs (Figure 5K). Treatment with BAI1 Ab:3DNA:Dox, but not the conjugate without doxorubicin, on the 1st and 13th days in culture eliminated Myo/Nog cells and myofibroblasts throughout the tissue and within a scratch wound one month after plating (Figure 5L,M). These studies illustrate that Myo/Nog cells are the progenitors of lens myofibroblasts in explants of human anterior lens tissue.

We also utilized the *human* anterior lens explant model to examine the response of Myo/Nog cells to dying cells [68]. Explants were labeled with a dye that remains within the plasma membrane and only fluoresces when phagocytosed into acidic compartments of other cells. The tissue was incubated with a high dose of free doxorubicin, dissociated and the cells were added to untreated explants. BAI1+ Myo/Nog cells engulfed prelabeled dead cells containing doxorubicin [68] (Figure 5N). 

Myo/Nog cells also phagocytose beads injected into the anterior chamber to induce glaucoma in *mice*, tattoo ink in human skin and dead cells in the retina [68,70]. Antibodies to markers of the monocyte lineage, ionized calcium binding adaptor molecule 1 (Iba1) and F4/80, did not bind to Myo/Nog cells in the skin, eyes or brain [16,61,68,72]. However, CD68, which is expressed in microglia, macrophages, other leukocytes, fibroblasts and endothelial cells [90,91], was found in a small subpopulation of BAI1+/Noggin+ cells in a *mouse* model of proliferative vitreoretinopathy (PVR) in a study described below [70]. This was not surprising because both CD68 and BAI1 are receptors for phosphatidylserine (PS) present in the outer leaflet of the plasma membrane of apoptotic cells [19,20,92,93]. We are unaware of reports showing that monocytes, macrophages or microglia express MyoD and Noggin, and expression of BAI1 in macrophages is controversial [94]. Therefore, we conclude that the Myo/Nog and monocyte lineages are distinct but have overlapping functions in clearance. 

## 4. Role of Myo/Nog Cells in Posterior Capsule Opacification

A fibrotic wound healing response, called posterior capsule opacification (PCO), occurs in approximately 20% of adults and most children within two years of cataract surgery [95,96,97,98,99,100]. PCO is characterized by the migration of cells and their differentiation to myofibroblasts whose contractions produce vision-impairing wrinkles in the posterior capsule [87,99,101,102]. We injected BAI1 Ab:3DNA:Dox into the *rabbit* lens during cataract surgery to test whether Myo/Nog cells are progenitors of myofibroblasts that cause PCO in vivo. In this aggressive model of PCO, a minimum of 80% of *rabbits* developed PCO within one month of surgery [66,103]. The drug dramatically reduced the number of myofibroblasts and wrinkles in the capsule without off-target effects [66] (Figure 6A–D). PCO, peripheral and anterior capsule opacification were reduced to below clinically significant levels [66]. This study established that Myo/Nog cells are the progenitors of myofibroblasts in the lens in vivo and revealed a potential therapeutic approach to preventing PCO. 

## 5. Additional Behaviors of Myo/Nog Cells in the Lens and Related Phenomena in Other Tissues

Having identified Myo/Nog cells as the progenitors of myofibroblasts in the lens, we turned our attention to PVR, a fibrotic disease that most often occurs in response to the repair of rhegmatogenous retinal detachment [104,105,106,107,108,109]. PVR is characterized by the formation of membranes on the inner surface of the retina (epiretinal), and/or behind or within the retina [104,105,106]. Myofibroblasts accumulate within the membranes and their contractions apply traction on the retina that may cause re-detachment [104,105,106,110]. In *human* epiretinal membranes, greater than 99% of cells with α-SMA, MyoD and striated muscle myosin II heavy chain co-expressed BAI1 and Noggin [67] (Figure 6F).

To examine the behavior of Myo/Nog cells during ERM formation, we utilized a *mouse* model in which PVR is induced by injecting SF_6_ gas and human retinal pigmented epithelial cells (ARPE-19) into the vitreous [111]. This model recapitulates disease progression seen in *humans* and other species. Myo/Nog cells were present on the inner surface of the retina early in the process of PVR induction and were the predominant cell type in ERMs [70]. They were the only cells that synthesized α-SMA apart from vascular smooth muscle and striated muscle myosin II [70]. Differentiated Myo/Nog cells overlaid retinal folds and areas of detachment (Figure 6G) and were present throughout the retina with disease progression (Figure 6H). Few leukocytes were present in the eye at all stages of PVR, indicating that inflammation did not appear to be a significant factor in Myo/Nog cell activation in this model. However, subpopulations of Myo/Nog cells expressed CD18 and CD45, markers for all leukocytes, and CD68 that is present in cells of the monocyte lineage [70]. We concluded that the death of *human* RPE cells was the likely stimulus for proliferation and migration of Myo/Nog cells and their engulfment of cell corpses [70], as occurs in other tissues, including the lens [68]. 

Although Myo/Nog cells are a distinct lineage in the lens, subpopulations of BAI1+ cells in *human* anterior lens explant cultures synthesized the beaded filament proteins filensin and CP49 that were considered specific markers of differentiated lens fibers [112,113,114,115]. Less than six percent of the total cells in the tissue fixed within 10 min of capsulorhexis were labeled with BAI1 or beaded filament antibodies [65]. Approximately 52% of filensin+ and 8% of CP49+ subpopulations contained BAI1 [65]. 

In parallel experiments, we were examining sarcomas with features of skeletal muscle for Myo/Nog cell markers. With the information in hand from lens tissue, we added filensin and CP49 to our screens for BAI1 and Noggin expression. Most cells expressing BAI1 and Noggin in cultures of *human* cell lines derived from alveolar and embryonal rhabdomyosarcoma (RMS) tumors contained beaded filaments [65] (Figure 7A–D). BAI1+/filensin+/CP49+ cells also were found in tissue sections from four subtypes of *human* RMS (Figure 7D–G) and Wilms tumors [65]. Leiomyosarcoma smooth muscle tumors, skin carcinomas, melanomas and normal skeletal muscle contained BAI1+/Noggin+ cells but they lacked detectable levels of filensin and CP49 [65]. These analyses revealed that beaded filament expression occurs outside of the lens and their localization in Myo/Nog-like cells appears to be diagnostic of tumors with features of skeletal muscle. 

## 6. The Road Ahead

During the past decade, we have focused primarily on testing potential therapeutic applications of depleting or adding Myo/Nog cells in the eye. However, basic questions related to the Myo/Nog lineage during the early stages of development remain unanswered. Although all three markers of Myo/Nog cells are identifiable in a small subpopulation in the blastocyst, it is unknown whether one or all of these molecules are present in the fertilized oocyte and sequestered with cell division or if their expression is induced regionally within the epiblast. Given the stability of expression of MyoD, Noggin and BAI1, even in the highly inductive environment of the developing embryo, it is reasonable to hypothesize that Myo/Nog cells of the adult are derived from founders in the early epiblast. 

Thus far we have not observed evidence for Myo/Nog cell multipotency, and therefore, these cells appear to be progenitors and not stem cells. Are they subject to mutations that promote tumorigenesis with aging, and if so, are they progenitors of RMS and Wilms sarcomas? In this regard, co-expression of Myo/Nog markers with filensin and CP49 within cultured explants of human lens tissue and tumors with features of skeletal muscle, but not other tumors or normal muscle, may provide insight into the regulation of beaded filament expression in Myo/Nog and differentiating LECs and possibly during transformation. 

Myo/Nog cell functions during lens development need to be further characterized. Exogenous or overexpressed Noggin delays lens fiber differentiation and expands the undifferentiated anterior lens epithelium [46,47,51,52]. A hypothesis worth testing is that the small population of Myo/Nog cells releases enough Noggin to balance LEC proliferation with differentiation and to maintain a population of lens fiber progenitor cells. Myo/Nog cells may also be involved in early stages of LEC induction, as suggested by studies in which *human* embryonic and induced pluripotent stem cells were treated with Noggin in the first stage of a three-step process to promote the development of lens progenitor cells and lentoid bodies [116,117,118,119,120].

Under homeostatic conditions, Myo/Nog cells are either slowly or non-dividing and synthesize Noggin without detectable translation of MyoD mRNA or expression of skeletal muscle proteins [9,13,15,29,30]. Several stimuli, including cell death, hypoxia, the presence of foreign material and wounding that occurs from surgical excision, abrasion, puncture and light damage, promote the expansion and migration of Myo/Nog cells [32,59,61,62,64,67,69,70,71,72]. The rapidity of their activation in adults was demonstrated in the lens in response to cataract surgery [66], the brain after focal injury [69] and in the skin where Myo/Nog cells emerge from their niche associated with hair follicles and populate the wound within 24 h of epidermal abrasion [61]. Paracrine factors that direct the migration of Myo/Nog cells are largely unknown but are predicted to be released from dying cells [121] and alter their adhesive properties as they become mobile and stationary again. The importance of Myo/Nog cell adhesion for differentiation was demonstrated in vitro by optimizing the composition of the extracellular matrix substrate to promote attachment, survival and formation of multinucleated skeletal myofibers [8,14,15]. 

Myo/Nog cell-derived myofibroblasts in PCO and PVR are, for the most part, single-nucleated cells that appear to lack sarcomeres. Skeletal myoblast fusion was shown to involve BAI1’s binding to PS on apoptotic cells in vitro and in vivo [20]. It is possible that BAI1+ myofibroblasts do not fuse due to a paucity of apoptotic cells in the aggregate. The absence of visible striations at the light microscopy level may indicate an absence of integrin beta-1 interactions with other membrane complexes and cytoskeletal elements that mediate sarcomere formation [122].

Contractions produced by Myo/Nog cell-derived myofibroblasts may be beneficial for wound closure in some tissues, but in the lens and retina, force generation perturbs architecture and impairs vision. Targeting Myo/Nog cells to prevent fibrotic disease in the lens is the most straightforward therapeutic application of our basic and translational research. The optimal anti-BAI1:3DNA:Dox concentration for depleting Myo/Nog cells and preventing PCO did not exhibit off-target effects when injected into the rabbit lens [66]. Reduction of Myo/Nog cells following cataract surgery is not expected to adversely affect LECs that remain after polishing and assist with securing the intraocular lens [123]. 

Myo/Nog cells are also a potential target for preventing PVR and retinal detachment [67,70]. Killing Myo/Nog cells in the normal retina did not appear to affect the tissue; however, in a model of oxygen-induced retinopathy, their depletion resulted in an increase in neuronal cell death [72]. Addition of Myo/Nog cells to the vitreous following severe light damage or into the brain following focal injury decreased cell death [63,69]. Therefore, the outcome of killing Myo/Nog cells to prevent PVR is expected to depend on the extent to which the retina is compromised at the time of intervention and whether enough of these cells remain or repopulate the tissue to mediate neuroprotection. The mechanism whereby Myo/Nog cells support neuronal viability is under investigation, and once identified, may lead to an approach that combines their depletion with the addition of a neuroprotective factor. 

In PCO and PVR, Myo/Nog cell-derived myofibroblasts continue to express BAI1 [66,67,70]. Targeting BAI1 to deliver a cytotoxin could conceivably be used as a non-surgical form of treatment, as well as preventing disease onset. Not all BAI1+/Noggin+ cells express muscle proteins in the fibrotic lens and retina, suggesting that a subpopulation undergoes self-renewal to maintain a pool of progenitor cells during active disease. 

Myofibroblasts also appear in other ocular diseases [87]. Myo/Nog cells are present throughout the eye, and therefore, may be a source of myofibroblasts in the cornea, as well as the lens and retina. Behaviors of Myo/Nog cells in glaucoma are also worthy of exploration. Related questions are whether they become activated in response to increased intraocular pressure and if their differentiation and contraction narrow the outflow passages. The phagocytic function of Myo/Nog cells highlights the importance of examining their role in maintaining the patency of the trabecular meshwork and Schlemm’s canal for outflow of the aqueous humor. In the lens, Myo/Nog cells’ appearance among lens fiber cells raises the possibility that they engage in phagocytosis that may be important for maintaining transparency. The relative contributions of Myo/Nog cells, macrophages and microglia to clearance in the eye are likely to vary between tissues and the extent of inflammation.

Myo/Nog cells’ behaviors of populating wounds and phagocytosis are shared with cells that mediate innate immunity. We characterized Myo/Nog cells as non-professional phagocytes to distinguish them from macrophages, microglia, neutrophils and epithelial cells (specialized phagocytes) that also engage in clearance [68]. Additional functions of Myo/Nog cells distinguish them from hematopoietic cell lineages, i.e., regulation of BMP signaling, skeletal myofiber formation and myofibroblast differentiation. Myo/Nog cells fit more broadly into the category of mediators of acquired resilience, defined as a stress-induced regulation of self-protective and self-repair mechanisms [124]. Phagocytosis, homing to wounds, differentiation to contractile myofibroblasts and mediation of neuroprotection are examples of Myo/Nog cells’ self-protective and repair behaviors. Our ongoing studies of the role of Myo/Nog cells’ production of BAI1 in regulating angiogenesis in the eye may identify another important self-protective mechanism to slow the progression of pathological vascularization. 

Our research has identified Myo/Nog cells as sole expressors of BAI1, MyoD and Noggin in the eye, mediators of phagocytosis and the source of myofibroblasts in the lens and retina. Myo/Nog cells are expected to be progenitors of myofibroblasts in other ocular tissues, as well as the heart, lung, kidney and liver whose functions are also compromised by fibrosis. Defining the molecular stimuli that activate Myo/Nog cells and trigger their proliferation, migration and differentiation may lead to additional therapeutic strategies to prevent and treat fibrotic disease. Their roles in regulating BMP signaling, neuroprotection and possibly angiogenesis could also be leveraged for therapeutic purposes. 

## Figures and Tables

**Figure 1 cells-12-01725-f001:**
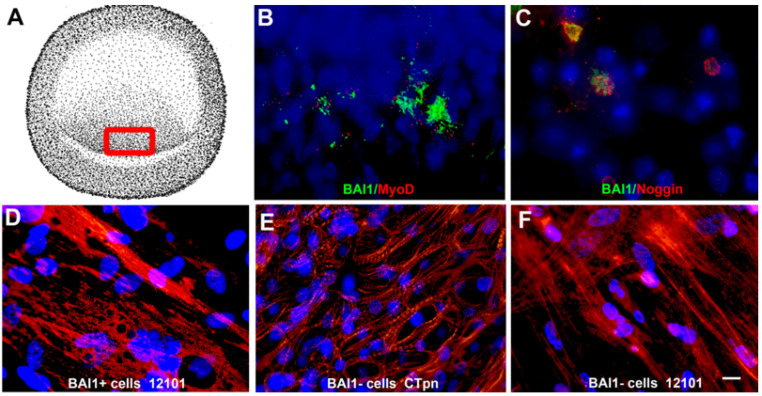
Cells that co-express MyoD, Noggin and BAI1 in the embryonic epiblast differentiate into skeletal muscle in vitro. In situ hybridization and immunofluorescence localization revealed a small population of cells that co-expressed BAI1 (green) and MyoD mRNA (red) (**B**) and Noggin protein (red) (**C**) in the posterior/medial epiblast of *chick* embryos before the onset of gastrulation (red box in **A**). Nuclei are stained with Hoechst dye (blue). Labeling with the skeletal muscle-specific 12,101 mAb demonstrated that BAI1-positive Myo/Nog cells isolated from the epiblast differentiated into skeletal myofibers when cultured in serum-free medium (**D**). BAI1-negative epiblast cells were labeled with a mAb to cardiac muscle-specific troponin T (**E**). Addition of Noggin to the medium resulted in the synthesis of the 12,101 antigen in BAI1-negative cells (**F**). Bar = 13 µM.

**Figure 2 cells-12-01725-f002:**
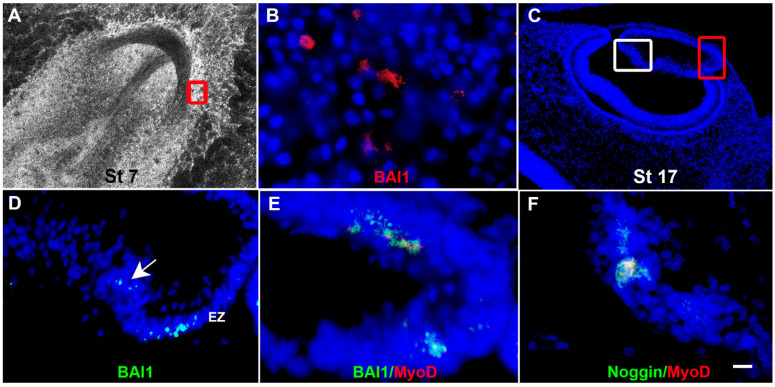
Epiblast-derived Myo/Nog cells are integrated into the developing lens. Myo/Nog cells were fluorescently labeled with the BAI1 mAb in the epiblast and the embryos were incubated for 1–3 days. Sections were stained with Hoechst dye to label nuclei (blue). In the stage 7 embryo, Myo/Nog cells were found in the pre-lens placode (boxed area in (**A**) shown at high magnification in (**B**)). A stage 17 eye is shown in (**C**). The area within the red box is shown at higher magnification in (**D**,**E**). The white box delineates the enlargement in (**F**). Prelabeled Myo/Nog cells were found in the equatorial zone (EZ) of the lens (**D**) and among the differentiating primary lens fiber cells (arrow in (**D**)). BAI1+ and Noggin+ cells (green) co-expressed MyoD mRNA (red) throughout the lens (**E**,**F**). Overlap of green and red appears yellow in merged images. Bar = 135 µM in (**A**), 13 µM in (**B**) and (**D**–**F**) and 56 µM in (**C**).

**Figure 3 cells-12-01725-f003:**
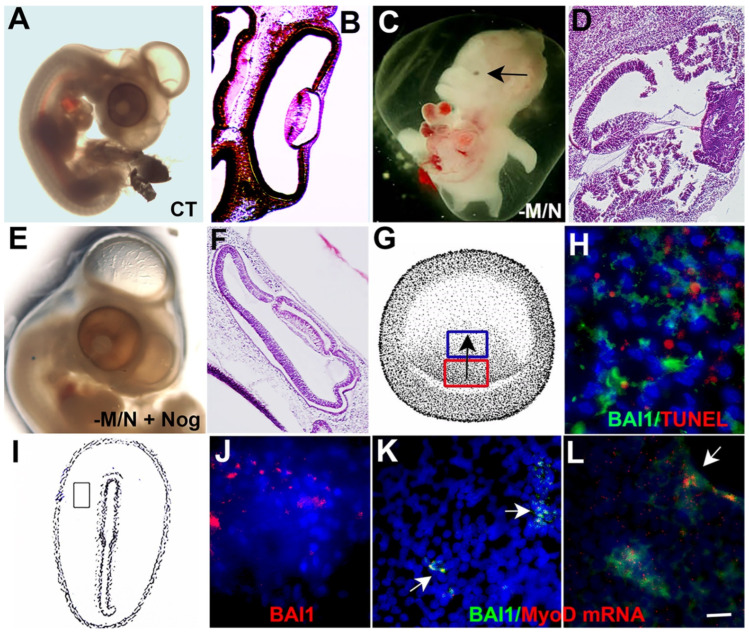
Myo/Nog cells are required for normal eye development and migrate to wounds in the embryo. Myo/Nog cells were eliminated in the *chick* embryo epiblast by incubating the embryo in the BAI1 mAb and complement. Embryos were incubated for approximately 5 days. Some embryos received beads soaked in Noggin (Nog) a day after treatment. A normal PBS control embryo (CT) and an H&E-stained section of the eye are shown in (**A**,**B**). Depletion of Myo/Nog cells resulted in malformations of the body wall and eyes, including microphthalmia (arrow in (**C**)) and dysgenesis of the lens and retina (**D**). Addition of Noggin-soaked beads prevented malformations caused by Myo/Nog cell depletion (**E**,**F**). Embryos treated with the D4 mAb and complement to eliminate a separate population of cells in the central epiblast (blue box in (**G**)) stimulated the migration of BAI1-labeled Myo/Nog cells (green) from the posterior/medial epiblast (red box in (**G**)) to TUNEL+ cells (red) (**H**). The direction of migration is indicated by the arrow in H. A puncture wound was produced in the day-1 chick embryo lateral to the primitive streak (boxed area in (**I**)). BAI1+ cells (red) surrounded the wound (**J**). BAI1+ (green) cells containing MyoD mRNA (red) were present in niches in explants of fetal chick lens tissue (arrows in (**K**)). Within two days after plating, BAI1+/MyoD mRNA+ cells had migrated to the cut edge of the tissue (arrow in (**L**)). Bar = 5 mM in (**A**,**C**), 56 µM in (**B**,**D**), 2.5 mM in (**E**), 41 µM in (**F**) and 13 µM in (**H**–**L**).

**Figure 4 cells-12-01725-f004:**
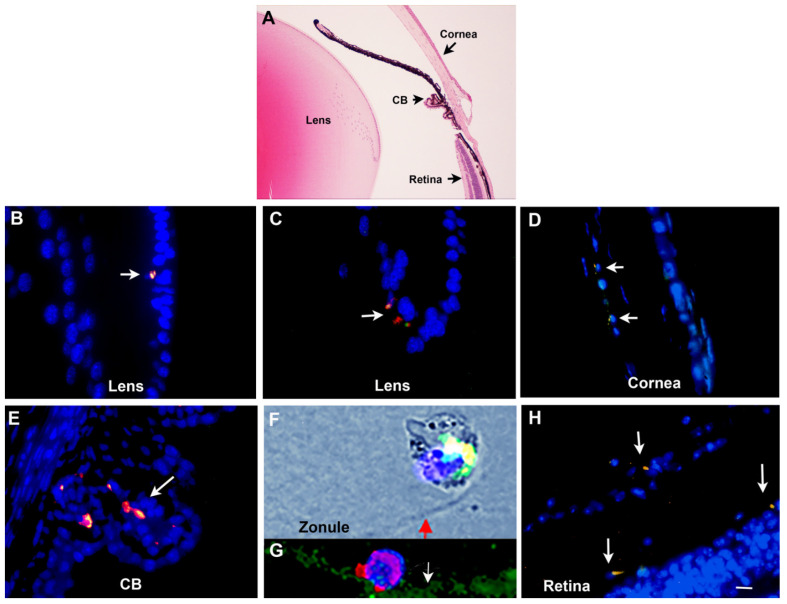
Myo/Nog cells are present throughout the adult *mouse* eye. Sections of the adult *mouse* eye were stained with H&E (**A**) or double-labeled with antibodies to BAI1 (red) and Noggin (**B**–**F**,**H**) or fibrillin (green) (**G**). The overlap of red and green appears yellow in merged images. Nuclei were stained with Hoechst dye (blue). Arrows point to Myo/Nog cells in the equatorial (**B**) and bow regions (**C**) of the lens, the endothelial layer of the cornea (**D**), ciliary body (CB) (**E**) and on the zonules of Zinn (arrows in (**F**,**G**)). Each layer of the retina contained a small number of Myo/Nog cells (**H**). Bar = 55 µM in (**A**), 9 µM in (**B**–**E**,**G**,**H**) and 6.5 µM in (**F**).

**Figure 5 cells-12-01725-f005:**
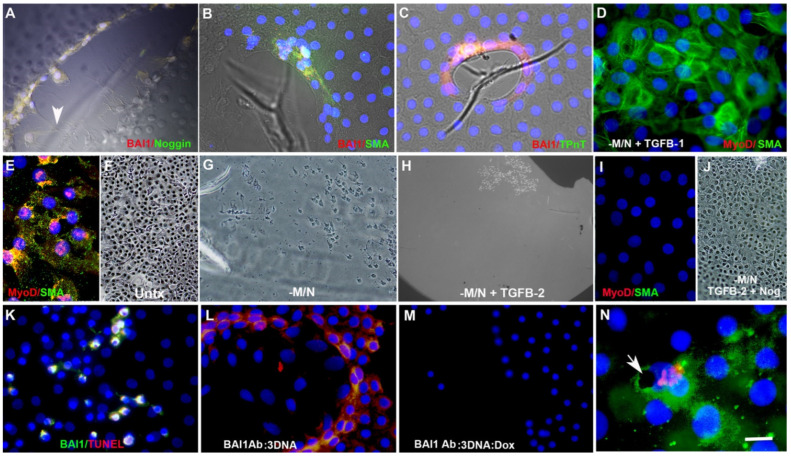
Myo/Nog cells are the source of myofibroblasts and phagocytose dead cells in human anterior lens explant cultures. Anterior lens tissue was obtained from patients undergoing cataract surgery and cultured in serum-free medium. Myo/Nog cells double-labeled with antibodies to BAI1 (red) and Noggin, α-SMA and skeletal muscle-specific troponin T (green) surrounded wounds in the epithelium (**A**–**C**) and extended processes towards wrinkles in the capsule (arrow in (**A**)). They also migrated onto the capsule and synthesized α-SMA after producing a scratch wound in 2-day cultures (**E**). Cells filling the scratch wound in the presence of Myo/Nog cells are shown in the phase contrast photomigraph (**F**). Myo/Nog cells were depleted with the anti-BAI1 mAb and complement (-M/N) (**G**–**J**). Treatment of explants with TGF-β1 after Myo/Nog cell depletion contained α-SMA+/BAI1-cells (**D**). Scratch wounds contained fewer cells on the capsule after Myo/Nog cell depletion (**G**). Incubation with TGF-β2 in the absence of Myo/Nog cells resulted in detachment of most cells from the capsule (**H**). Addition of Noggin (Nog) to TGF-β2 treated cultures prevented detachment and promoted migration into the wound, but the cells did not express MyoD or α-SMA (**I**,**J**). Explants were treated on days 2 and 13 with the BAI1 mAb conjugated to 3DNA nanocarriers intercalated with doxorubicin (BAI1 Ab:3DNA:Dox) (**K**,**M**). TUNEL staining revealed apoptotic BAI1+ cells (**K**). BAI1+/α-SMA+ cells migrated into the scratch wound in 30-day control cultures treated with BAI1 Ab:3DNA lacking doxorubicin (**L**). BAI1+/Noggin+ were not present in explants treated with BAI1 Ab:3DNA:Dox (**M**). Anterior lens explants were labeled with lysosensor red dye and then treated with doxorubicin to induce apoptosis. Treated cells were added to untreated explant cultures. BAI1+ cells (green) phagocytosed cells prelabeled with lysosenser dye (red) and killed with doxorubicin (black) (arrow in **N**). Bar = 9 µM in (**A**–**E**,**I**,**K**–**N**), 27 µM in (**F**–**I**).

**Figure 6 cells-12-01725-f006:**
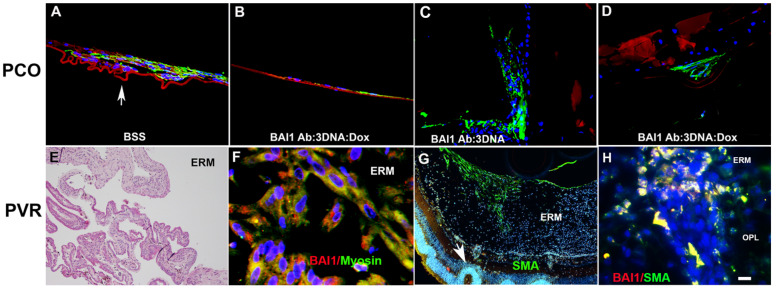
Myo/Nog cells differentiate into myofibroblasts in posterior capsule opacification (PCO) and proliferative vitreoretinopathy (PVR). Rabbit lenses were injected with balanced saline solution (BSS) (**A**,**C**) or BAI1 Ab:3DNA:Dox (**B**,**D**) during cataract surgery. Lenses were harvested 30 days later and sections were labeled with an antibody to SMA (green) (**A**–**D**). SMA and wrinkles (arrow) were abundant in lenses injected with BSS) (**A**). Depletion of Myo/Nog cells significantly reduced α-SMA+ myofibroblasts and wrinkles in the capsule (**B**,**D**). Human epiretinal membranes (ERM) removed from patients with PVR were sectioned and stained with H&E (**E**) or antibodies to BAI1 (red) and striated muscle myosin (green) (**F**). Overlap of red and green in merged images appears yellow. Nuclei were labeled with Hoechst dye (blue). Myosin+ myofibroblasts express BAI1 in human epiretinal membranes (**F**). PVR was induced in the mouse retina by injecting SF6 gas and human ARPE-19 cells into the vitreous. Sections were double labeled with antibodies to BAI1 (red) and α-SMA (green) (**G**,**H**). BAI1+ myofibroblasts were present in epiretinal membranes and throughout the retina (**G**,**H**) and overlaid areas of detachment (arrow in **G**). OPL = outer plexiform layer. Bar = 27 µM in (**A**–**D**), 9 µM in (**E**–**H**) and 135 µM in (**G**).

**Figure 7 cells-12-01725-f007:**
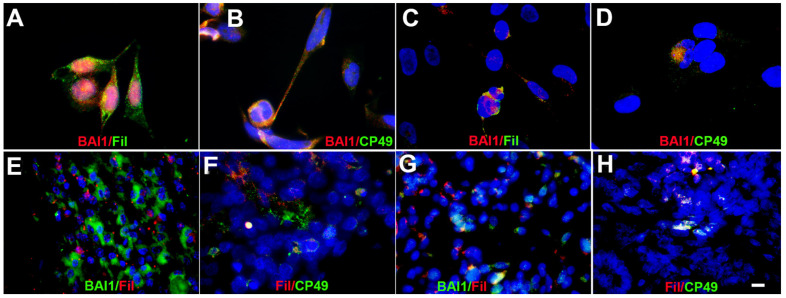
BAI1+ cells in *human* rhabdomyosarcoma cell lines and tumors express beaded filament proteins. Cultures of RC13 (**A**,**B**) and RD RMS cell lines (**C**,**D**) derived from alveolar and embryonal RMS tumors, respectively, were double labeled with antibodies to BAI1 (red) and filensin (Fil) or CP49 (green). Nuclei were stained with Hoechst dye (blue). Both cell lines contained cells that co-expressed BAI1 and beaded filaments (**A**–**D**). Tissue sections of alveolar (**D**,**E**) and embryonal RMS tumors (**F**,**G**) were double-labeled with the BAI1 mAb (green) and an anti-sense probe for filensin mRNA (red) (**D**) or an antibody to filensin (green) (**D**,**F**). Double labels also were performed with antibodies to filensin (red) and CP49 (green) (**E**,**G**). BAI1+ cells expressed filensin and filensin co-localized with CP49 (**D**–**H**). Bar = 9 µM.

## Data Availability

The RC13 and RD rhabdomyosarcoma cell lines were obtained from the American Type Culture Collection.

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
