# Peer review of "Myo/Nog Cells: The Jekylls and Hydes of the Lens"

_cells, 2023, doi:10.3390/cells12131725_

Round 1

Reviewer 1 Report

The manuscript entitled “Myo/Nog Cells: The Jekylls and Hydes of the Lens” by Gerhart and George-Weinstein reviews the information currently known about a novel cell type that they discovered and named Myo/Nog cells, based upon their co-expression of MyoD-mRNA, Noggin protein and BAI1, with respect to the lens. The authors also review a recent paper they published in IOVS concerning Myo/Nog cells giving rise to myofibroblasts in a mouse model of proliferative vitreoretinopathy (PVR).

The manuscript does a good job of describing Myo/Nog cells and reviewing their origin and presence in ocular tissues. The authors also review their published evidence that these cells are the source of myofibroblasts that cause posterior capsular opacification and contribute to the pathology of PVR.

My comments mostly relate to the figures and lack of agreement between the figures and the text.

1.       Figure 1 – missing “D” in the figure.

2.       Figure 1 in panel F, I believe that the label should read “BAI1- cells with Noggin 12101

3.       Figure 2: the authors should point out what BAL1 positive cells are in primary fibers in D. Most of the signals in D appear to be in the epithelium.  Are the green signals in the fiber region really cells?  In the legend, “BAI1+ cells (green) co-expressed MyoD and Noggin (red) throughout the lens (E and F)” According to the figure, BAL1 is green in 2E and Noggin is green in 2F while MyoD is red in both 2E and 2F.  The figure legend is confusing here.

4.       In lines 99-101: “Eye malformations ranged from anophthalmia to lens dysgenesis and overgrowth of the retina [29,31] (Fig. 3A and B). Treatment with the G8 mAb or complement alone did not impair eye development (Fig. 3 C and D).” Fig. 3A and B refer to the control eye and there are no figures showing the controls treated with the G8 mAb or complement alone.  Fig. 3C and D show the effects of the elimination of the Myo/Nog cells. The text should be reworded to more clearly refer to what is being shown in the figure.

5.       In lines 136-140 of the text, the references to the figure are incorrect. I believe that the text is referring to figure 3 I-L, but instead the text is talking about Fig. 4A and B, Fig. 4C and Fig. 3D. This needs to be corrected.

6.       In figure 5, it is unclear what the difference between E and F is.  While E is obviously immunofluorescence with antibodies to MyoD and αSMA, is F the same region in brightfield?  What does Untx stand for?  This abbreviation is undefined.  It is also unclear where the scratch wound is/was relative to the uninjured tissue in both F and J.

Author Response

The authors thank you for your attention to inconsistencies and inaccuracies in the legends and the text describing the figures. We apologize for the errors. The following corrections and changes were made to the paper.

D was added to Figure 1.

The annotation in Figure 1F was corrected.

An arrow was added to the cluster of Myo/Nog cells residing among the primary fiber cells in Fig. 2D.

The only green fluorescence in the tissue section shown in Fig. 2D was from Myo/Nog cells that were pre-labeled with anti-BAI1 antibody and a fluorescein conjugated secondary antibody while they resided in the epiblast and integrated into the developing lens (Gerhart et al., 2009, Dev. Biol. 336, 30).

The probe for MyoD mRNA fluoresced red and the primary antibodies were tagged with green in Fig. 2E and F. This was clarified in the legend.

The text in lines 99-101 and legend for Fig. 3 were corrected and clarified. The control eyes displayed in Fig. 3A and B were embryos treated with PBS. Images of embryos treated with the G8 mAb or complement alone were not included in the figure and are now described as “not shown”. These control experiments are described in Gerhart et al., 2009, Dev. Biol. 336, 30. 

The references to the photographs of the puncture wound in the embryo (Fig. 3J) and the fetal lens Fig. 3K and L) were corrected in the text.

Fig. 5 E and F are photographs of cells that had populated the wound within 48 hours of scratching the epithelium. The photographs are different magnifications of different explant cultures. Untreated means that explants were not incubated in anti-BAI1:3DNA:Dox to kill Myo/Nog cells and were not treated with TGF-ß after producing a scratch wound. The explants shown in E-J all received a scratch wound. The methodology was clarified in the figure legend.  

Reviewer 2 Report

The authors reported that Myo/Nog cells are the only expressing cells of BAI1, MyoD and Noggin in the eye, mediators of phagocytosis and a source of myofibroblasts in the lens and retina. Furthermore, by identifying the molecular stimuli that activate Myo/Nog cells and cause their proliferation, migration and differentiation, the paper proposed a potential new therapeutic strategy for the prevention and treatment of fibrotic diseases.

The review article is of great interest and the content is well organized. The reviewer comment on the following points, but expect serious corrections from the authors

1. The Noggin described in the manuscript has also been used in the differentiation of iPS cells into lens cells and several papers have been reported. The reviewer urge you to add one additional sentence in “The road ahead section”, with references cited.

2. As human samples are used, the name of the ethics committee that reviewed the study and the approval number should be added in the Research Ethics section.

3. Add scale bars to all figures. This corrective action must be carried out.

4. In Figure 2C, explain the areas enclosed by white and red lines. Do the red squares indicate Figure 2D and the white squares indicate Figure 2F? Which area is Figure 2E an enlarged photograph?

5. Figure 3b is of significantly poorer quality than the other photographs. Re-take or replace the photograph. Figures 3B, 3D and 3F must be photographs of the same magnification.

6. The photograph in Fig. 4A is not white balanced. Re-take or replace the photograph.

Author Response

The authors thank the reviewer for commenting on finding the manuscript interesting and well organized. The following changes were made to the paper.

A sentence describing the use of Noggin to promote lens differentiation from embryonic stem cells and iPS cells, and references was added to the third paragraph of the road ahead section.

A sentence was added to the Research Ethics sections stating that the studies involving human tissue were approved by the Institutional Review Board of the Philadelphia College of Osteopathic Medicine.

Scale bars and equivalent microns were added to the figures and legends.

Sentences were added to the legend of Figure 2 indicating that the area shown in the red box in C is shown at higher magnification in D and E, and the white box region is shown in F.

The sections from this experiment were from 2008 and could not be rephotographed. We added an alternative for Fig. 3B, although it is more heavily stained than the other H&Es. It does display morphology.

The image in Fig. 3F was enlarged to more closely match those of B and D.

The color of Figure 4A was adjusted.